# Evaluating the Effectiveness of Educational Intervention on ICU Nurses’ Knowledge of Delirium: A Quasi-Experimental Approach

**DOI:** 10.3390/nursrep15060205

**Published:** 2025-06-06

**Authors:** Jamal Qaddumi, Khaled Awawdi, Mahdi Tarabeih

**Affiliations:** 1Department of Nephrology, An-Najah National University Hospital, Omar Ibn Al-Khattab St., Nablus P.O. Box 7, Palestine; jamal9877@najah.edu; 2Department of Public Health, Faculty of Medicine and Health Sciences, An-Najah National University, Nablus P.O. Box 7, Palestine; awawdi.h@iac.ac.il

**Keywords:** ICU delirium, critical care nurses, educational program, CAM-ICU, patient outcomes

## Abstract

**Background and Objective:** Delirium, a prevalent neurocognitive disorder, frequently affects critically ill patients hospitalized in intensive care units (ICUs), leading to increased mortality, prolonged hospital stays, and higher healthcare costs. This quasi-experimental study assessed the effect of an educational program relating to ICU nurses’ knowledge of delirium in a university hospital in Nablus, Palestinian Authority. **Methods:** A pre-test–post-test design was employed, utilizing a 25-item questionnaire for 114 ICU nurses. The educational intervention included a presentation on delirium, the distribution of educational materials, and follow-up video sessions. Our study aim was to evaluate nurses’ understanding of ICU delirium and the impact of the educational program on their ability to identify and evaluate the delirium. **Results**: Pre-intervention assessments indicated limited awareness among nurses regarding delirium diagnosis and management tools, i.e., the Confusion Assessment Method for the ICU (CAM-ICU) and the Intensive Care Delirium Screening Checklist. Post-intervention results showed a significant improvement in knowledge; median scores increased from 6 (range: 3–13) to 15 (range: 12–20) (*p* < 0.001). Nurses also reported greater confidence in identifying and managing delirium, and 50% found CAM-ICU easy to use. However, knowledge gaps remained concerning mixed delirium types and modifiable risk factors. **Conclusions:** Continuous educational programs are essential for ensuring long-term knowledge retention. We recommend integrating routine delirium education with hospital policies and emphasizing the use of delirium assessment tools during each shift. Findings show that targeted education can enhance ICU nurses’ competencies and thereby improve patient outcomes through more effective delirium management.

## 1. Introduction

Intensive care unit (ICUs) create an environment that can either trigger or worsen psychological disorders or neuropsychiatric issues such as post-traumatic stress disorder [1] anxiety, and depression [2]. Delirium is one of the most common behavioral manifestations following an ICU stay [3]. Estimates have suggested that delirium may occur in over half of all hospital admissions [4]. The prevalence of delirium can vary due to inconsistent terminology, often being interchangeably referred to as acute brain dysfunction, septic encephalopathy, or ICU syndrome [5]. According to the Diagnostic and Statistical Manual of Mental Disorders (DSM)-5, delirium is a disturbance in attention and awareness that develops over a short period of time (usually hours to a few days) and tends to fluctuate in severity throughout the day. The primary cause is believed to be a direct physiological consequence of a medical condition, medication, or an intoxicating agent [6]. ICU delirium is a major concern in critically ill patients as it can significantly impact their outcomes and the course of their illness [7].

Delirium presents in different motoric phenotypes: hyperactive, hypoactive, and a mixed type [8]. Major risk factors for developing delirium in the ICU include the need for mechanical ventilation, the severity of illness, and advanced age [9]. While there is no single unifying mechanism for delirium, various functional, neurotransmitter, and injury-related hypotheses have been proposed. Accurate diagnosis requires the use of validated diagnostic tools [10].

Delirium is characterized by fluctuating levels of consciousness, impaired attention, cognition, and disrupted sleep–wake cycles [11,12]. It must be emphasized that delirium in the ICU is a critical concern, with its incidence ranging from 32% to 87%. Patients on mechanical ventilation for medical or surgical procedures experience higher rates at 60–80% [13,14]. Delirium is associated with increased risks of mortality, infection, readmission, prolonged hospital stays, and elevated hospitalization costs, all of which exacerbate the related medical and social burdens [15]. Moreover, family members are significantly impacted, with 75% experiencing anxiety due to the patient’s condition [16]. Recently, guidelines have recommended the use of non-pharmacological interventions, which have gained considerable attention [17,18].

The ICU is a potentially pathogenic environment for critically ill patients who are at risk of mental disturbances due to additional distressing factors, including metabolic disturbances, restrictions, and environmental factors (i.e., light and noise), that disrupt the patient’s natural sleep cycle and can lead to mental disorders including delirium [19]. Among the acute physiological and mental state changes to which ICU patients are vulnerable, delirium is particularly concerning as it is associated with dementia onset as well as long-term cognitive decline, which might be delayed by the implementation of delirium prevention strategies [20].

Based on psychomotor behavior, delirium can be divided into three subtypes: hyperactive, hypoactive, and mixed delirium. Whereas the type most frequently identified outside of the ICU is hyperactive delirium, hypoactive and mixed types are more frequently seen in the ICU [21]. Patients suffering from hypoactive delirium may have worse outcomes than those with hyperactive delirium [19].

ICU delirium with hyperactivity represents ~23% of cases, characterized by agitation, restlessness, emotional instability, and positive psychotic traits such as hallucinations (that frequently obstruct the provision of care). Symptoms of hypoactive delirium usually include confusion, sedation, apathy, decreased responsiveness, slowed motor function, a withdrawn attitude, lethargy, and sleepiness. The most typical type, ~50% of all cases, is mixed delirium, exhibiting a fluctuation between hypoactive and hyperactive features [21].

Risk factors of delirium are classified into predisposing factors and potentially modifiable/precipitating factors. The most prevalent predisposing factors are aging, dementia, functional disabilities, male gender, poor vision, hearing loss, and a mild cognitive ailment. Precipitating factors vary. For instance, medications such as psychoactive or cholinergic medications affect up to 39% of delirium cases. Other precipitating factors are operations, anesthesia, untreated pain, infections, hypoxia, acute illnesses, and the acute aggravation of chronic illnesses [22]. These precipitating factors originate in the ICU, whereas the predisposing factors are already present prior to the patient’s admission. The difficulty in defining delirium lies in the fact that predisposing and precipitating risk factors interact with one another in multitudinous ways that influence the effects and outcomes [23].

Due to delirium’s fluctuating course and the complexity of ICU care, diagnosis is challenging; therefore, doctors and researchers may combine results from many tests [24].

The Confusion Assessment Method for the Intensive Care Unit (CAM-ICU) and the Intensive Care Delirium Screening Checklist (ICDSC), recommended by Critical Care Medicine (CCM), are the most commonly employed screening instruments [10,25,26]. Only patients who respond to voice commands can be assessed for delirium; however, since most ICU patients are sedated due to mechanical ventilation, a scale is required to measure sedation or impaired consciousness; two such scales are the Richmond Agitation and Sedation Scale (RASS) and the Sedation and Agitation Scale (SAS) [27].

Delirium in ICU patients increases mortality, prolongs ICU stays, delaying treatment for other patients spreads illness, and raises patient costs; thus, it is very important for ICU nurses to acquire a knowledge of delirium and have the ability to diagnose and evaluate delirium [20,28].

An intervention study examining the efficacy of delivering an educational program to improve the nurses’ knowledge and the screening of delirium found significant correlation between the nurses’ newfound knowledge and the number of screened ICU patients with delirium [20,29].

### Study Premises and Hypotheses

This study is based on the following premises:ICU delirium is a common, underdiagnosed condition that negatively affects patient outcomes.Critical care nurses play a central role in the early recognition and management of ICU delirium.Targeted educational interventions can improve nurses’ knowledge and confidence in identifying and managing delirium.

Based on these premises, the following hypotheses were proposed:**H1:** ICU nurses will demonstrate a statistically significant improvement in knowledge scores regarding ICU delirium after an educational program.**H2:** There will be increased awareness and use of formal delirium assessment tools (e.g., CAM-ICU) following the intervention.**H3:** Nurses’ post-intervention knowledge scores will not differ significantly based on demographic characteristics such as age, gender, or education level.

## 2. Materials and Methods

### 2.1. Study Design

This was a quasi-experimental, one-group pre-test–post-test design conducted in ICU settings across three hospitals. The study focused on assessing the critical care understanding of ICU delirium and evaluating the short-term impact of a structured educational program on the critical care nurses’ knowledge of ICU delirium.

The educational intervention was designed with SMART (Specific, Measurable, Achievable, Relevant, Time-bound) objectives to assess its effectiveness. The primary objective could be described as follows:

Specific: We sought to increase ICU nurses’ knowledge about delirium, including its types, risk factors, diagnostic criteria, and the use of assessment tools such as the CAM-ICU.

Measurable: Effectiveness was measured by comparing pre-test and post-test knowledge scores using a validated 25-item questionnaire. A statistically significant increase in median knowledge scores was considered an indicator of effectiveness.

Achievable: The intervention included a focused lecture, handouts, and video training, which were tailored to ICU nurses’ baseline knowledge and workload.

Relevant: The program targeted critical knowledge gaps in ICU delirium recognition and management, which were directly related to improving clinical practice.

Time-bound: Knowledge improvement was evaluated within one week post-intervention through structured post-test assessment administration.

To evaluate the effectiveness of the delirium educational program, this study utilized a modified version of a knowledge assessment scale, adapted from previous studies [30,31,32]. The tool consisted of a demographic section: gender, age, and education level. This was followed by 21 multiple-choice questions, assessing knowledge related to delirium in the intensive care unit, and an additional 4 questions, specifically evaluating delirium using the CAM-ICU tool.

In the initial phase, a pilot study was conducted to examine the reliability of the tool. Informed consent was obtained from participating ICU nurses who voluntarily agreed to take part in the study. A paper-based knowledge questionnaire was distributed as a pre-test. Subsequently, a focused 45 min educational session was conducted, which included a comprehensive overview of delirium, its definition, clinical significance, pathophysiology, risk factors, types, clinical implications for ICU patients, and an introduction to delirium assessment tools and management strategies.

The post-training knowledge questionnaire was administered two weeks after the session. The completion time for both the pre-program and post-program questionnaires was approximately 20 min. The tool demonstrated acceptable internal consistency, with a Cronbach’s alpha of 0.76 for all 25 items.

### 2.2. Research Question: PICO Framework

This study was structured around the PICO framework to define the research question:

Population (*p*): Critical care nurses working in ICUs in hospitals in Nablus, Palestinian Authority. Intervention (I): Implementation of a structured educational program on ICU delirium. Comparison (C): Knowledge levels of ICU nurses before the educational intervention (pre-test). Outcome (O): Improvement in ICU nurses’ knowledge and awareness of delirium, including use of diagnostic tools (post-test).

Based on this structure, the research question was as follows:

“Among ICU nurses, does participation in an educational program on ICU delirium, compared to baseline knowledge, lead to improved understanding and ability to identify and manage delirium?”

### 2.3. Sample Population and Sampling

The population comprised all nurses working in the ICU. A non-probability convenience method was used to choose the participants due to logistical and ethical constraints in the ICU setting, where the randomized selection of participants was not feasible. The sample size was determined using the G-power version 3.1 program [17]. We used statistical analysis via a paired-sample *t*-test with an effect size of 0.5, an alpha error probability of 0.05, and 1 beta probability of 0.95. This allowed us to calculate the sample size: (114) critical care nurses. The sample size calculation was based on the parameters suitable for a paired-sample *t*-test (effect size = 0.5, α = 0.05, power = 0.95), which yielded a minimum requirement of 45 participants. The final sample size of 114 participants was not solely based on an attrition adjustment. Rather, the number reflects the total number of eligible and available ICU nurses across all participating hospitals during the study period. No participant dropout was recorded; thus, the final sample included all nurses who met inclusion criteria and completed both the pre- and post-program tests.

This study employed a non-probability convenience sampling method due to logistical and ethical constraints in the ICU setting, where the randomized selection of participants was not feasible. We acknowledge that convenience sampling may limit generalizability and introduce sampling bias. However, it is commonly accepted in quasi-experimental healthcare research when randomization is impractical.

This study only included registered nurses who had spent more than one year providing care in the coronary care unit (CCU), surgical intensive care unit (SICU), and medical intensive care unit (MICU). We excluded nurses who were on leave or those who did not complete both the pre- and post-program tests.

### 2.4. Site and Setting

The study took place in Nablus hospitals: the Al-Watani hospital maintains an ICU and CCU ward; the An-Najah National University Hospital is a specialized hospital treating patients in the SICU, MICU, ICU, and CCU wards; and Al-Arabi is a specialized hospital maintaining ICU and CCU wards.

### 2.5. Data Collection Procedure

Upon obtaining the required ethical approval, the researchers visited each facility mentioned above, informing the staff of the study and assisting them in assigning research facilitators who would approach eligible participants and explain the study to them. Those who expressed interest in participating were contacted by the researchers who explained the study’s purpose and its significance, and assured the participants that their participation in the study would pose no direct or indirect harm to them.

### 2.6. Validity and Reliability

The content validity of this questionnaire was reviewed by 7 experts: 4 faculty doctors and 3 experienced nurses from the ICU. The input provided by these specialists was taken into account when updating the questionnaire’s content. For the 25 items listed, the tool’s internal consistency reliability of 0.80 was shown to be adequate.

### 2.7. Statistical Analysis

Data were not normally distributed when the Kolmogorov–Smirnov normality test was run on the SPSS version 25 after data had been coded, entered, and analyzed. Chi-square analysis determined the median and range for the descriptive data. By using the Mann–Whitney U test, the comparison of the overall knowledge score between the pre- and post-program test was examined.

### 2.8. Ethical Consideration

The research was carried out according to the guidelines set forth in the Declaration of Helsinki. Permission was granted by the Institutional Review Board (IRB) of the An-Najah National University. Confidentiality and anonymity were ensured by not recording the participant’s names, and only numbering them in a sequential series for data analysis. Data were obtained from the participants for research purposes only. Participation was completely voluntary. It was also explained to the participants that they could withdraw from the study at any time without giving a reason, and that they could see the completed study’s results.

### 2.9. Educational Program Content and Structure

The educational program was developed based on international guidelines for ICU delirium recognition and the best educational practices for adult learners. The program aimed to enhance ICU nurses’ knowledge and confidence in identifying and managing delirium using evidence-based tools.

Program Content:Overview of ICU delirium: definition, types (hyperactive, hypoactive, mixed), and clinical significance;Risk factors: predisposing vs. precipitating;Diagnostic tools: in-depth training on the CAM-ICU and ICDSC;Consequences of delirium on patient outcomes;Case-based discussions and simulated clinical vignettes (video format);Introduction to non-pharmacological interventions for delirium prevention.

Structure and Delivery Format:Lecture (60 min): Delivered in-person in the hospital seminar conference room by expert faculty, using PowerPoint slides and clinical examples.Self-directed Learning (60 min): Nurses received printed educational handouts and guidelines for review at their own pace.Video Session (30 min): A recorded session demonstrating the use of CAM-ICU in real clinical scenarios, made available via the hospital’s online platform.

Total Duration: This was approximately 2.5 h. The materials were delivered over a single day to ensure feasibility and minimal disruption to ICU workflows. The program was reviewed and approved by ICU specialists and senior nurse educators to ensure clinical and pedagogical relevance.

Given the operational limitations of ICU staffing and time constraints, this educational program was designed as a concise, theory-based intervention focused on knowledge improvement rather than hands-on skill acquisition. The primary aim was to enhance the cognitive understanding of ICU delirium—its identification, subtypes, risk factors, and the use of screening tools.

Due to institutional and scheduling constraints, practical sessions such as role-play or bedside simulation could not be incorporated. Consequently, the assessment of learning outcomes relied on a pre- and post-intervention questionnaire, which focused on knowledge recall, the interpretation of clinical scenarios, and correct use of delirium screening tools.

## 3. Results

### 3.1. Results Related to Demographic Variables

The Kolmogorov–Smirnov normality test was run; data were not normally distributed. The median and range of the chi-square test were used to analyze the descriptive data. The Mann–Whitney U and the Kruskal–Wallis tests compared the overall knowledge scores for the pre- and post-intervention tests to determine whether there was a correlation between the demographic characteristics of the nurses and their level of knowledge. The study included 114 ICU nurses; the majority were males (71.9%) between the ages of 20 and 29 (54.4%). Most of the nurses held a BA (73.7%) (Table 1).

### 3.2. Results Related to the Research Question and Hypothesis

To examine the effect of the education program on ICU delirium knowledge, the median of the pre-test score was compared with post-test score of nurses’ knowledge regarding ICU delirium. The results showed a quantum leap in the nurses’ knowledge regarding delirium in the post-test score (median = 15) compared to the pre-test score (median = 6), (significant *p* value < 0.001 (Table 2).

Most responses saw a significant rise in post-test nurses’ ICU delirium knowledge scores. In post-test interventions, approximately 60% of the nurses responded to the questions regarding their assessment of the patient’s altered level of consciousness compared to 10.5% in the pre-test design (*p* < 0.001); 56.1% correctly answered the feedback questions regarding the incidence of delirium compared to 5.3% in the pre-test (*p*< 0.001). Although there was a statistically significant increase in correct answers, three questions, unfortunately, had a relatively low percentage of correct answers in the post-test scores. Delirium can occur in up to 56.1% of patients. When assessing for an altered level of consciousness, it is important to stop the assessment for RASS (57.9%) and modifiable risk factors (39.6%).

Furthermore, 77.1% of the nurses correctly described the features of delirium with a high significance (*p* < 0.001); 73.7% properly answered the feedback of hypoactive delirium with a marginally significant level (*p* = 0.058) when compared with the pre-test scores (47.4%). The participants showed a higher level of knowledge regarding signs of delirium in the post-test than in pre-test (73.7% vs. 35.1%, respectively, *p* < 0.001). Knowledge of the impact of delirium on costs and the mortality of patients following hospital discharge was found to be associated with high knowledge scores, ranging from 63.2% to 70.2% when compared to the pre-test scores, at a significant level (*p* < 0.001). Finally, the post-test analysis, for the question regarding increased costs and mortality, found a huge increase in the knowledge score compared with the pre-test results (Table 3). The percentages shown in Table 3 refer specifically to correct responses. The *p*-values revealed whether the improvement was statistically significant, indicating that the educational program had a measurable impact on nurses’ knowledge.

Based on the findings, the study’s three hypotheses were confirmed:

H1 was corroborated: ICU nurses demonstrated a statistically significant improvement in knowledge scores after the educational program. The median score increased from 6 (range: 3–13) to 15 (range: 12–20), with a *p*-value < 0.001, indicating robust knowledge gains following the intervention (Table 2).

H2 was corroborated: There was a marked increase in awareness and the use of formal delirium assessment tools, especially the CAM-ICU. Post-intervention, >50% of nurses found the CAM-ICU easy to use, and 50.9% reported that it should be used at least once per shift, thus demonstrating increased tool adoption (Table 3).

H3 was corroborated: Post-intervention knowledge scores did not significantly differ based on demographic characteristics such as age, gender, or education level (*p* > 0.05 across all variables in Table 4), supporting the hypothesis that the intervention was equally effective across diverse nurse subgroups.

According to Table 4, there are small differences in the nurses’ knowledge scores regarding ICU delirium, attributed to age and education level; nurses >50 years old and those with a diploma had the lowest knowledge scores regarding delirium (3.5 and 5, respectively). However, these variations between the scores of nurses’ knowledge as to ICU delirium and their demographic data exhibited no statistical significance.

According to Table 5, the nurses’ perception of the importance of the ICU delirium assessment tool was 72%, whereas 19.3% considered it an essential tool. Unfortunately, nearly half (44%) of the nurses were unaware of the existence of an assessment tool for ICU delirium. Only 29.8% of the nurses claimed that they had heard about ICDSC; however, 26.3% never used the tool. Over 56% of the nurses were aware of an assessment tool for ICU delirium. Only 29.8% of the nurses claimed that they heard about the CAM-ICU, and just 10.6% of the nurses stated that they had used this tool over the last month.

Almost a third (31.5%) of the participants believed that the delirium tool was very important; 61.5% believed that it was essential. This reflects the improvement in the nurses’ perception of the importance of the delirium tool. Over half (52.6%) of the participants were quite confident in their evaluation, whereas 22.8% were quite confident based on the post-awareness confidence scores obtained from the CAM assessments. ICU nurses believed that using the CAM is typically very easy (28.1%) or quite easy (56.1%). Given the use of the CAM-ICU, 50.9% of the participants indicated that assessments must be conducted at least once every shift (Table 6).

## 4. Discussion

Research has shown that the majority of ICU nurses lack a knowledge of ICU delirium, a severe condition that affects critically ill patients. The goal of the current study was to evaluate how well an educational program can succeed in increasing the knowledge of ICU nurses regarding patients with ICU delirium. Compared to the 38.1% to 66% values reported in previous studies, the critical care nurses’ knowledge at the pre-intervention stage yielded a comparatively low median score (28.5%) [20,33,34]. The knowledge score of our nurses was similar to those observed in other studies [35,36,37].

The instrument employed to measure ICU delirium knowledge varied between research studies, which could account for the small variations in the results [32,33,34]. Furthermore, only critical care nursing master’s degree courses address ICU delirium; diploma programs do not. Consequently, it is possible that nurses without master’s degrees were less knowledgeable about this condition [38]. In other words, nurses may see delirious patients in the ICU, yet, they may not have received evidence-based training or the necessary information. Nonetheless, the results show a noteworthy improvement in critical care nurses’ understanding of delirium and the delirium assessment tool following an interventional education program, with an overall knowledge of 71.5%. This increase was independent of the nurses’ demographic characteristics and is in accordance with earlier research [20,33,37,38].

Irrespective of the nurses’ educational background, we showed that critical care nurses’ understanding of ICU delirium and the delirium assessment instrument may be considerably increased with a quick and easy educational intervention [16]. However, other research studies have demonstrated that a comprehensive, long-term educational program affects nurses’ knowledge in the long run [32,39,40] and that the scores for most of the questions increased as much as in our work. For example, 70.2% of the participants correctly answered the question as to the impact of ICU delirium; 66.7% of the participants correctly answered the question regarding the effect of ICU delirium on critically ill patients and the hospital. According to a 2023 study, the score for certain items did not considerably increase after the intervention. Only 49% of the participants correctly answered the question regarding the impact of ICU delirium; ~20% of the participants accurately responded to the question about the effects of ICU delirium on critically ill patients.

For the questions addressing different subtypes of delirium and how to differentiate between them, >50% of the participants answered the questions correctly after the intervention. An additional item that was not substantially altered after the intervention was the ICU delirium subtype, in contrast to the research performed by Gesin et al. [32]. Contrary to a 2008 study, the authors found that nurses are better at recognizing hyperactive delirium than hypoactive delirium, which might give rise to the myth that hyperactive delirium poses a greater risk to patient safety and should therefore be more worrisome. These findings were similar to those of various studies [20,32,37,38,39,40] performed during the past few years. The most effective way to educate medical professionals regarding ICU delirium is bedside teaching, which has been shown to increase nurses’ understanding. Furthermore, a study on the proficiency and knowledge of critical care nurses in delirium assessment found that improving critical care nurses’ understanding of delirium requires adequate education [41]. We found that scores regarding the three criteria indicated that conventional education programs may not be entirely effective; it is crucial to consider that the post-test assessment had fewer questions. Traditional educational programs must, therefore, fit with pedagogically sound practices, such as bedside instruction.

The results of earlier research provide insight into the significance of educational programs [19,32], demonstrating an inability to identify and prevent delirium. According to this study, the educational program assisted in raising the nurses’ overall knowledge of delirium and screening from 28.5% to 71.5%, in agreement with [40] After a brief educational intervention, research studies found a positive correlation between ICU delirium screening and delirium knowledge. A delirium diagnosis and evaluation cannot occur without an ongoing educational program [18,19,20,32,39,40]. After the interventional program, nurses’ comprehension of ICU delirium significantly improved [38]. A similar study [20] was conducted at a Canadian urban tertiary care institution’s ICU and it found that teaching led to an improvement in perceived tool usefulness, and ultimately perceptions of physician value and utility. Nonetheless, the authors also pointed out that without constant work, knowledge development is fleeting.

Our findings, however, indicate that both highly skilled academic nurses and those with only diplomas demonstrated a comparable level of knowledge acquisition. The teaching program usually enhances the critical care nurses’ understanding of ICU delirium, which may increase their confidence in managing delirious patients. However, there was a drop in knowledge ratings in three areas: mixed delirium, modifiable risk factors, and risk factors for developing delirium. It is plausible that the participants encountered difficulties in comprehending the material pertaining to delirium risk factors or that the issue was not adequately communicated throughout the training session. Further research and development are required.

### 4.1. Limitation

A relatively small sample size was chosen, which may affect the generalizability of the findings to the broader population of ICU nurses. We assessed the impact of an educational intervention in the short- term, immediately after the training. This approach may not capture a long-term retention of knowledge and its application in clinical practice. The reliance on self-reported data from the nurses regarding their knowledge and practices could introduce bias as participants might overestimate their understanding or the effectiveness of the intervention. The participants possessed varying levels of education and experience, which might have influenced the outcomes. The intervention may have produced different levels of effectiveness depending on the participants’ baseline knowledge and experience. Furthermore, no practical or simulation-based assessments were included, limiting the evaluation to theoretical knowledge improvement only.

These limitations suggest that while the study provides valuable insights, further research with larger sample sizes, follow-ups audits, bedside assessments, and OSCE-style evaluations is needed to validate and expand upon our findings.

### 4.2. Recommendations

We recommend that future educational programs aimed at enhancing critical care nurses’ understanding of ICU delirium should include the following key strategies:Routine Education Programs: Given the high prevalence and incidence of delirium amongst critically ill patients, combined with the low levels of nurse knowledge observed, we recommend that the Palestinian Authority’s Ministry of Health implement routine educational courses and programs focusing on delirium.Policy Changes in Hospitals: Hospitals should revise their policies to ensure that nurses apply delirium assessment tools, i.e., the (RASS and the CAM-ICU), at least once per shift. This change would increase the nurses’ awareness and early detection of ICU delirium.Continuous Education: Ongoing educational programs are crucial for maintaining and improving the nurses’ knowledge as to ICU delirium. Knowledge gained from short-term educational interventions is often temporary,; therefore, continuous efforts are necessary to ensure long-term retention and application.Tailored Teaching Methods: Educational programs should be customized to address the specific needs and knowledge gaps of critical care nurses. This should focus on areas where nurses demonstrated lower knowledge scores, i.e., mixed delirium and risk factors for delirium. An emphasis on teaching methods, such as bedside instruction, has been proven to increase nurses’ understanding.Comprehensive Training: More extensive training sessions, including practical and hands-on learning opportunities, should be integrated into the programs. This approach is more effective in increasing nurses’ confidence and competence in diagnosing and managing ICU delirium.Assessment and Feedback: Regular assessments of nurses’ knowledge and skills related to ICU delirium should be conducted in order to identify areas of improvement. Feedback from these assessments should be used to continually adjust and enhance educational programs.Further Research: Researchers should conduct additional studies to identify the most effective methods for reducing delirium amongst critically ill patients and increasing nurses’ knowledge of delirium. These recommendations aim to create a more effective learning environment for nurses, ultimately leading to improved patient outcomes in the ICU.

## 5. Conclusions

This study demonstrated a statistically significant improvement in ICU nurses’ knowledge of delirium following a targeted educational program, with median knowledge scores rising from 6 to 15 (*p* < 0.001). Post-intervention, there was also a substantial increase in awareness and the self-reported usage of delirium assessment tools, particularly the CAM-ICU. These findings confirm that even brief, structured educational interventions can yield meaningful improvements in theoretical knowledge and perceived competence among critical care nurses.

## Figures and Tables

**Table 1 nursrep-15-00205-t001:** Demographic characteristics of participants (Total N = 114).

Demographic Data	N	Percent (%)
Age in Years		
20–29	62	54.4%
30–39	42	36.8%
40–49	6	5.3%
50–59	4	3.5%
Gender		
Male	82	71.9%
Female	32	28.1%
Education Level		
Diploma	12	10.5%
BA (Bachelor’s Degree)	84	73.7%
MA (Master’s Degree)	18	15.8%

**Table 2 nursrep-15-00205-t002:** Median and range score of nurses’ knowledge pre- and post-test.

Test	Median	Range	*p*-Value
Pre-test	6	3–13	Mann–Whitney U
Post-test	15	12–20	*p* < 0.001

**Table 3 nursrep-15-00205-t003:** Comparison between nurses pre- and post-test scores for each question about delirium knowledge.

Question	Pre-Test (N = 114)	Post-Test (N = 114)	*p* Value
1. Delirium can occur in up to:			
a. 10% of the ICU patients b. 33% of the ICU patients c. 67% of the ICU patients d. 83% of the ICU patients	3 (5.3%)	32 (56.1%)	<0.001
2. Delirium can present as:			
a. Hyperactive symptoms b. Hypoactive symptoms c. Mixed (i.e., both hyperactive and hypoactive symptoms) d. All of the above	20 (35.1%)	42 (73.7%)	<0.001
3. The following term can be used interchangeably with delirium:			
a. Sun downing b. ICU syndrome c. ICU psychosis d. None of the above	27 (47.4%)	44 (77.1%)	0.002
4. Characteristics of delirium include:			
a. Disturbances of consciousness b. A change in cognition or development of perceptual disturbances c. A condition that occurs over the course of several weeks d. A and B	22 (38.6%)	44 (77.1%)	<0.001
5. All the following are true of mixed delirium except:			
a. It has symptoms of hyperactive delirium b. It has symptoms of hypoactive delirium c. It is more common than pure hyperactive delirium d. It is more common than pure hypoactive delirium e. All the above	30 (52.6%)	51 (89.5%)	<0.001
6. Delirium has been associated with all of the following except for:			
a. Increased mortality b. Increased incidence of coma c. Increased length of stay d. Higher costs of care e. 3 times greater re-intubation rate	13 (22.8%)	38 (66.7%)	<0.001
7. Medications associated as risk factors for delirium include:			
a. Antipsychotics (i.e., Haldol) b. Anticholinergic (i.e., Cogentin) c. Corticosteroids (i.e., Solumedrol) d. Benzodiazepines (i.e., Versed) e. B, C, and D	20 (35.1%)	43 (75.4%)	<0.001
8. When assessing for an altered level of consciousness it is important to:			
a. Consider the patient’s level of consciousness over an entire shift b. Stop the assessment for a RASS of −3 to −5 c. Consider the effect of recently administered sedation or analgesia therapy on your patient’s level of consciousness d. All of the above	6 (10.5%)	33 (57.9%)	<0.001
9. Characteristics of delirium include:			
a. Development over a short course of time (hours to days) b. The course is characterized by fluctuations c. The course is progressive d. A and B e. B and C	27 (47.4%)	41 (71.9%)	0.013
10. Hyperactive delirium is characterized by:			
a. Agitation b. Withdrawal c. Flat affect d. A and B e. A and C	17 (29.8%)	43 (75.4%)	<0.001
11. Hypoactive delirium is not characterized by all, except:			
a. Withdrawal b. Apathy c. Lethargy d. Decreased responsiveness e. Agitation	20 (35.1%)	41 (71.9%)	<0.001
12. The most common type of delirium seen in the ICU is:			
a. Hyperactive delirium b. Hypoactive delirium c. Mixed delirium d. A and C are equally common	19 (33.3%)	42 (73.7%)	<0.001
13. Hypoactive delirium is characterized by:			
a. Withdrawal b. Restlessness c. Flat affect d. Apathy e. A, C and D	27 (47.4%)	42 (73.7%)	0.058
14. Ordering the subtypes of delirium from least to most common:			
a. Hyperactive, hypoactive, mixed b. Hypoactive, hyperactive, mixed c. Hypoactive, mixed, hyperactive d. All are equally common	19 (33.3%)	41 (71.1%)	<0.001
15. During an ICU stay, delirium is associated with all of the following except for:			
a. Increased mortality b. Development of multi-organ dysfunction c. Increased length of stay d. 3 times greater re-intubation rate e. Higher costs of care	13 (22.8%)	36 (63.2%)	<0.001
16. After hospital discharge, delirium is associated with:			
a. Requirement for care in chronic care facility b. Decreased functional status at 6 months c. No long-term sequelae d. A and B	23 (40.4%)	37 (64.9%)	0.004
17. All of the following are risk factors for the development of delirium except for:			
a. Increased age b. Disruptive family/visitors c. Use of physical restraints d. Use of tubes and catheters	16 (28.1%)	41 (71.9%)	<0.001
18. After hospital discharge, delirium is associated with:			
a. Increased mortality b. Development of dementia c. Long-term cognitive impairment d. B and C e. All of the above	12 (21.1%)	40 (70.2%)	<0.001
19. All of the following regarding increased costs and mortality for patients with delirium are true except for:			
a. Occurs if delirium was ever present b. Increases with each additional day spent in delirium c. Can result in an average of $10,000+ in costs d. Can result in a 3 times mortality risk e. All of the above statements are true f. Lack of nutrition	2 (3.5%)	38 (66.7%)	<0.001
20. Modifiable risk factors for delirium include all of the following except for:			
a. Lack of daylight/clocks/orienting items b. A noisy environment c. Sepsis d. Baseline pulmonary disease e. All are modifiable risk factors	1 (1.8%)	34 (39.6%)	<0.001
21. The difference between hallucinations and delusions is that:			
a. Hallucinations are the perception of something that is not there with no stimulus and delusions are false beliefs that are fixed/ unchanging b. Delusions are the perception of something that is not there c. No stimulus and hallucinations are false beliefs that are fixed/unchanging d. The terms are interchangeable. e. Hallucinations are the perception of something that is not there f. A known stimulus and delusions are false beliefs that are fixed/unchanging	16 (28.1%)	47 (82.5%)	<0.001

Assessment tool reprinted with permission from Ref. [30]. 2023, Zahra Sameer Aldawood et al.

**Table 4 nursrep-15-00205-t004:** Association between overall delirium knowledge scores and nurses’ demographic characteristics.

Demographics Data	Median (Range)	*p*-Value
**Age in years**		KW = 0.162
20–29	7 (3–12)	
30–39	6 (3–13)	*p* = 0.300
40–49	8 (5–8)	
50–59	3.5 (3–4)	
**Gender**		MW = 0.720
Male	6 (3–13)	*p* = 0.963
Female	6 (5–12)	
**Education**		KW = 0.128
Diploma	5 (3–11)	
BA	6 (3–12)	*p* = 0.164
MA	8 (3–13)	

KW = Kruskal–Wallis, MW = Mann–Whitney U.

**Table 5 nursrep-15-00205-t005:** Nurses’ perception as to the importance of the delirium assessment tool (pre-test).

Question	N	Percent (%)
How important is assessing delirium?		
Not important	0	0%
Important	5	8.8%
Very important	41	71.9%
Essential	11	19.3%
Have you heard of any formal test of delirium applicable to the ICU?		
Yes	32	56.1%
No	25	43.9%
Which test have you heard of?		
CAM-ICU (Confusion Assessment Method of the ICU)	17	29.8%
ICDSC (Intensive Care Delirium Screening Checklist)	17	29.8%
DDS (Delirium Detection Scale)	2	3.5%
NuDESC (Nursing Delirium Screening Checklist)	1	1.8%
Have not heard of any test	20	35.1%
Did you use a formal test of delirium when you made your assessment over the last month?		
Always	6	10.6%
Often	11	19.3%
Sometimes	20	35.1%
Rarely	5	8.8%
Never	15	26.3%

Assessment tool reprinted with permission from Ref. [30]. 2023, Zahra Sameer Aldawood et al.

**Table 6 nursrep-15-00205-t006:** Nurses’ perception regarding the importance of a delirium assessment tool (Post-test).

Nurses’ Perception About the Importance of Delirium Assessment Tool (Post-Test)	N	Percent
How important is assessing delirium?		
Not important	0	0%
Somewhat important	4	7%
Very important	18	31.5%
Essential	35	61.5%
Are you confident your assessment using the CAM-ICU was accurate (compared to a full psychiatric assessment of delirium, or your own usual assessment of a patient mental state)?		
Very confident	13	22.8%
Quite confident	30	52.6%
Not very confident	13	22.8%
Not at all confident	1	1.8%
Did you think that the CAM-ICU was easy to use?		
Yes, usually very easy	16	28.1%
Yes, usually quite easy	32	56.1%
No, usually quite hard	9	15.8%
No, usually very hard	0	0%
How often should bedside nurses make this assessment?		
Once a day	3	5.3%
Twice a day	9	15.8%
Once per shift	29	50.9%
More than once per shift	16	28.1%
Are you confident your assessment using the CAM-ICU was accurate? (compared to a full psychiatric assessment of delirium, or your own usual assessment of a patient mental state)		
How important is assessing delirium?		
Not important	0	0%
Somewhat important	4	7%
Very important	18	31.5%
Essential	35	61.5%
Are you confident your assessment using the CAM-ICU was accurate? (compared to a full psychiatric assessment of delirium, or your own usual assessment of a patient mental state)		
Very confident	13	22.8%
Quite confident	30	52.6%
Not very confident	13	22.8%
Not at all confident	1	1.8%
Did you think the CAM-ICU was easy to use?		
Yes, usually very easy	16	28.1%
Yes, usually quite easy	32	56.1%
No, usually quite hard	9	15.8%
No, usually very hard	0	0%
How often should bedside nurses make this assessment?		
Once a day	3	5.3%
Twice a day	9	15.8%
Once per shift	29	50.9%
More than once per shift	16	28.1%

Assessment tool reprinted with permission from Ref. [30]. 2023, Zahra Sameer Aldawood et al.

## Data Availability

Individual-level data cannot be made publicly available due to legal and ethical restrictions. Aggregative data might be provided upon reasonable request to the corresponding author.

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
