# Peer review of "Evaluating the Effectiveness of Educational Intervention on ICU Nurses’ Knowledge of Delirium: A Quasi-Experimental Approach"

_nursrep, 2025, doi:10.3390/nursrep15060205_

Round 1
Reviewer 1 Report
Comments and Suggestions for Authors
- Please fill in the abbreviations in your table: BA and MA
- Page 5: in the chapter of result you are writing: "The study included 114 ICU nurses". please correct the sum of included ICU nurses or the numbers in your tables. The numbers in your table do not represent this number. In the tabel their is a report of 150 nurses.
- Please explain in detail what the results in table 3 are demonstrating to the reader. For example: Question No:1. Delirium can occur in up to:
a. 10% of the ICU patients
b. 33% of the ICU patients
c. 67% of the ICU patients
d. 83% of the ICU patients
What is the message to the reader about the results of 3 (5.3%) and 32 (56.1%) and <0.001* ? does only 3 nurses answer the question correctly in the pre test and 32 nurses answer the question after the intervention in the post-test? this might not be clear to the reader. therefore we recommended to explain this issue in detail please.
Reviewer 2 Report
Comments and Suggestions for Authors
Thank you for the opportunity to review this manuscript titled "Impact of an Educational Program on Critical Care Nurses’ Knowledge of ICU Delirium: A Quasi-Experimental Study." The topic is important and relevant to critical care practice. I offer the following comments for the authors’ consideration:
Introduction:
The content across pages 2 and 3 appears repetitive. Consider consolidating overlapping information to improve clarity and flow.
Methods – Sampling and Sample Size Calculation:
The manuscript indicates that a non-probability convenience sampling method was used, yet reports a sample size calculation based on parameters appropriate for inferential statistics (paired-sample t-test). This raises methodological concerns, as such calculations assume random sampling to ensure statistical validity. Furthermore, the reported sample size calculation (n=45) and the adjusted sample size (n=114) do not align with the stated 15% attrition rate. Based on the figures provided, the adjustment appears inconsistent and warrants clarification.
Educational Intervention:
More detail is needed regarding the educational program itself. Information such as its content, structure, delivery format, and duration should be clearly described in the main text rather than limited to a table, to support replicability and contextual understanding.
Reviewer 3 Report
Comments and Suggestions for Authors
Dear Authors,
The topic's significance, complexity, and associated complications, costs, and mortality certainly justify the need for ICU personnel education in this field
Several areas require revision, clarification, restructuring, and further development.
- The abstract is excessively lengthy and convoluted, containing superfluous detail; restructuring is necessary.
- Similarly, the introduction is overly extensive, presenting a disorganized sequence of definitions, risk factors, further definitions, descriptive and evolutionary aspects, and consequences of delirium. Improved clarity and a more coherent framework are needed through restructuring and systematization.
- The study's hypotheses and premises should be explicitly stated.
- PICO questions should be systematically addressed.
- As an educational program, we must formulate SMART educational objectives. For example, to address ,,the effectiveness of the educational program", please define specific criteria for measurement.
- The methodology need detail about the study type, participant inclusion/exclusion criteria, questionnaire types, program characteristics (curriculum, time allocation for theoretical, individual/video study, and educational resources—online platform, printed materials, lectures), instructor team structure, and a roadmap encompassing summative, formative, and certifying assessments, long-term knowledge retention, and follow-up on practical application skills.
- Given the absence of practical sessions and reliance solely on questionnaires for participant evaluation, the tailored educational strategies must be clarified.
- The conclusions section must be revised to reflect the study's findings. For instance, we need to substantiate how the educational program could potentially influence policy changes based on the data collected.
Round 2
Reviewer 3 Report
Comments and Suggestions for Authors
Dear authors,
Thank you for your thoughtful responses and for considering my suggestions.
The content and presentation of the work have been significantly enhanced. It is clear that a retrospective with prospective value study of an educational program presents numerous challenges.
I believe the next step should involve case simulations, structured communication training, and the integration of bedside sessions into the program's structure.